# Comparative Transcriptomics for Genes Related to Berberine and Berbamine Biosynthesis in Berberidaceae

**DOI:** 10.3390/plants11202676

**Published:** 2022-10-11

**Authors:** Neha Samir Roy, Nam-Il Park, Nam-Soo Kim, Yeri Park, Bo-Yun Kim, Young-Dong Kim, Ju-Kyung Yu, Yong-In Kim, Taeyoung Um, Soonok Kim, Ik-Young Choi

**Affiliations:** 1Agriculture and Life Sciences Research Institute, Kangwon National University, Chuncheon 24341, Korea; 2Department of Plant Science, Gangneung-Wonju National University, Gangneung 25457, Korea; 3NBIT, Kangwon National University, Gangwondaehakgil-1, Bodeumkwan 504, Chuncheon 24341, Korea; 4Plant Resources Division, National Institute of Biological Resources, Incheon 22689, Korea; 5Department of Life Science, Multidisciplinary Genome Institute, Hallym University, Chuncheon 24252, Korea; 6Syngenta Crop Protection LLC, 9 Davis Drive, Research Triangle Park, NC 27709, USA; 7On Biological Resource Research Institute, Chuncheon 24239, Korea; 8Microorganism Resources Division, National Institute of Biological Resources, Incheon 22689, Korea

**Keywords:** berberine, berbamine, *Berberis*, comparative transcriptomes, methyltransferase, cytochrome P450 mono-oxidase

## Abstract

Berberine and berbamine are bioactive compounds of benzylisoquinoline alkaloids (BIAs) present in *Berberis* species. The contents of berbamine are 20 times higher than berberine in leaf tissues in three closely related species: *Berberis koreana, B. thunbergii* and *B. amurensis*. This is the first report on the quantification of berberine compared to the berbamine in the *Berberis* species. Comparative transcriptome analyses were carried out with mRNAs from the leaf tissues of the three-species. The comparison of the transcriptomes of *B. thunbergii* and *B. amurensis* to those of *B. koreana, B. thunbergii* showed a consistently higher number of differentially expressed genes than *B. amurensis* in KEGG and DEG analyses. All genes encoding enzymes involved in berberine synthesis were identified and their expressions were variable among the three species. There was a single copy of CYP80A/berbamunine synthase in *B. koreana*. Methyltransferases and cytochrome P450 mono-oxidases (CYPs) are key enzymes for BIA biosynthesis. The current report contains the copy numbers and other genomic characteristics of the methyltransferases and CYPs in *Berberis* species. Thus, the contents of the current research are valuable for molecular characterization for the medicinal utilization of the *Berberis* species.

## 1. Introduction

Berberidaceae is an herbaceous plant family that consists of 700 species of 18 genera [1,2]. The family includes pharmacologically important genera (i.e., *Berberis, Caulophyllum, Mahonia*) producing bioactive compounds such as alkaloids, terpenoids, flavonoids, sterols, anthocyanins, and carotenoids [3,4]. Berberidaceae are commonly known as the barberry family because the majority of the species are in the genus *Berberis,* in which about 500 species are included [5]. *Berberis* species are distributed worldwide in temperate and subtropical regions of the northern hemisphere [6]. Plants in the *Berberis* species have been utilized for folk medicine for a long time. Extracts from many parts of the organs have been used for treatments in wide range of cardiovascular, metabolic, hepatic, and renal disorders [7]. The bioactive compounds in *Berberis* species are benzylisoquinoline alkaloids (BIAs). BIAs are restricted to certain plant families such as Magnoliaceae, Ranunculaceae, Papaveraceae, and Berberidaceae in Ranunculales [8] and Fabaceae in Fabales [9]. BIAs are nitrogen-containing plant secondary metabolites with about 2500 molecules, including morphine, codeine, sanguinarine, papaverine, etc. [10,11]. Berberine, a nonbasic quaternary benzylisoquinoline alkaloid, is the most biologically active BIA compound distributed in almost all *Berberis* species [3,12]. Berberine is important in medicinal chemistry because it is involved in the modification of functional groups in strategic positions in designing novel and selective drugs [7,13]. The bioactive and medicinal effects of berberine are cholagogic, hepatoprotective, sedative, antibacterial, antioxidant, anti-inflammatory, anticonvulsive, anticholinergic, etc. [12,14]. Berbamine, a natural bis-benzylisoquinoine alkaloid (bisBIA), is also a major bioactive component found in *Berberis* species [15,16]. Berbamine has been attracted by various researchers for the antitumor activities in various types of tumors [15,17,18].

The pathways for BIA biosynthesis have been well established by combined analyses of genomics and metabolomics [11,19,20]. The BIA biosynthesis pathway starts with two L-tyrosine molecules that are converted to dopamine and 4-hydrophenylacetaldehyde. These two molecules are combined to become (*S*)-norcoclaurine, an immediate precursor to all BIA compounds. Hagel et al. (2015) [11] performed transcriptome analyses in 20 taxonomically related species in four families within Ranunculales that produce various BIA compounds in which they established the functional homologs and the catalysts within BIA metabolism. Liu et al. (2017) [19] isolated whole sequences of 16 metabolic genes for sanguinarine and chelerythrine of BIAs in *Macleaya cordata* in Papaveraceae from complete genome sequencing. The plants in the genus *Coptis* in Ranunculaceae produce various BIAs. He et al. (2018) [20] identified 53 and 52 unigenes for BIA biosynthetic pathways in *Coptis teeta* and *C. chinensis*, respectively, from a comprehensive transcriptome study and posited the biosynthetic pathways for various BIAs including berberine. Metabolome profiling in combination with the ultra-performance liquid chromatography-electrospray ionization tandem mass spectrophotometry (UHPLC-ESI-MS.MS) analysis retrieved 64 full-length transcripts encoding the compounds putatively involved in BIA biosynthesis in *Coptis deltoidei* [21]. Beyond the Ranunculales, lotus (*Nelumbo nucifera*) in Nelumbonaceae, Proteales produce various compounds of BIAs [22]. Deng et al. (2018) [9] investigated the BIA biosynthetic pathway and its transcriptional regulation by RNA-Seq analysis using the Illumina Hiseq platform in which the aporphine alkaloids in lotus leaves resulted from (*S*)-N-methylcoclaurine precursor instead of the (*S*)-reticuline precursor common for other BIAs [20,21].

Compared to the numerous studies of BIAs, the biosynthetic pathway for bisBIAs was poorly investigated. The *N*-methylcoclaurines were precursors for bisBIAs in the feeding analysis of ^14^C-labeled tyrosine, tyramine, and several chiral 1-benzyl-1,2,3,4-tetrahydroisoquinolines in the cell culture of *Berberis stolonifera* [23]. The *N*-methylcoclaurines arise from the methylation of coclaurine that is catalyzed by norcoclaurine 6-*O*-methyltransferase (6OMT) [24].

The pharmacological effects of the berberine from *Berberis* species are available from many studies [25,26,27]. However, reports on the gene expression or genomic studies for berberine synthesis are relatively poor compared to other species that produce BIAs [19,20,21]. (*S*)-tetrahydroprotoberberine oxidase (STOX), an enzyme that functions in catalyzing (*S*)-canadine to berberine, was purified from suspension culture of *Berberis wilsoniae* [28], and the enzymatic activity of the recombinant STOX was confirmed by heterologous expression in *Spodoptera frugiperda* insect cells [29]. Roy et al. (2021) [30] identified 16 genes encoding the enzymes in berberine synthesis among 23,246 full-length unigenes by transcriptome analysis in *Berberis koreana* using the PacBio sequencing platform. They demonstrated that the berberine synthesis was streamlined by the absence of genes for enzymes of other BIAs but not by the presence of all the genes for berberine biosynthesis in *B. koreana*. The PacBio sequencing results revealed that gene duplication and translation into isoforms might contribute to the functional specificity of the duplicated genes and isoforms in plant alkaloid synthesis. The current report contains comparative transcriptomic analyses in three *Berberis* species (*B. koreana, B. amurensis*, and *B. thunbergii*) in Berberidaceae to elucidate the berberine and berbamine biosynthesis pathway.

## 2. Materials and Methods

### 2.1. HPLC Analysis

Young leaves were air-dried and ground to fine powder with mortar and pestle. BIAs were extracted from dried power samples with 70% ethanol (1 g/10 mL) in dark for 24 h at room temperature. Then the extract was filtered through a 0.22 μm syringe filter, and the extract was diluted to 10,000 mg/L for HPLC analysis. The analysis of berbamine and berberine was conducted using a Shimadzu Prominence HPLC system (Shimadzu, Kyoto, Japan) equipped with a diode array UV-vis detector for monitoring at 280 nm. Compounds were separated using a C18 column (250 × 4.6 mm, 5 μm, Waters, Ireland). Binary gradient elution was performed using solvent A (water containing 0.1% formic acid) and solvent B (acetonitrile containing 0.1% formic acid), which were delivered at a flow rate of 1.0 mL/min as follows: 0 min, 0% B; 5 min, 0% B; 10 min, 30% B; 25 min, 40% B; 30 min, 50% B; 35 min, 0% B; and 40 min, 0% B. The injection volume was 10 μL, and the column temperature was 40 °C.

### 2.2. RNA Extraction, Illumina Sequencing, Data Processing and Assembly

All samples were collected in August 2021 when plants grow most vigorously in Chuncheon, South Korea. Leaf tissues were collected from three *Berberis* species (*B. koreana*, *B. amurensis,* and *B. thunbergii*) and immediately frozen in liquid nitrogen. Three biological replicates for each leaf species were prepared. RNA extraction was performed with Hybrid- RTM kit (Gene All Biotechnology Co., Seoul, Korea) with the manufacturer’s protocol. RNA quality and quantity were checked using the A2100 Bio system (Agilent Technologies Inc., Santa Clara, CA, USA). Then, only RNA with a concentration of 1–10 μg and an RNA integrity number (RIN) above 8.0 was used for library construction for sequencing.

A cDNA library was constructed using the TruSeq RNA Sample Prep Kit v2 (Illumina). Paired-end sequencing was performed with Illumina HiSeq 2500 at the National Instrumentation Centre for Environmental Management (NICEM) at Seoul National University, South Korea. Low-quality reads with Phred score < 20, short reads (<50 bp), empty nucleotides (N at the end of reads), and adapter sequences were trimmed using Trimmomatic software [30]. Quality control before and after trimming was performed using FastQC [31] (https://www.bioinformatics.babra ham.ac.uk/projects/fastqc/; access date, 11 May 2022). 

### 2.3. De Novo Assembly

Obtained sequences were assembled using the de novo RNA-Seq Assembly Pipeline (DRAP) protocol [32]. Only high quality clean reads were used in the assembly using Oases program. The first step for assembly using Oases (v 0.2.09) [33] was implemented with K mers sizes of 25, 31, 37, 43, and 49, and the assembled fragments were merged. Then, the chimeric contigs were deleted and we removed duplicated sequences, insertions, and deletion and polyA/T tails. The second compaction employed with the protocol of CD-HIT-EST v4.8.1 to order the contigs by length and remove all the included ones given identity and coverage thresholds [34]. The coding region was then finally confirmed using TransDecoder (v3.0.1) [35]. Short reads were mapped to the assembly using Burrows-Wheeler Aligner (BWA) [36]. Fragments Per Kilobase per Million (FPKM) was calculated to select fragments of higher than 1 FPKM value to be used as reference sequence. HISAT2 v2.1.0 program [37] was used for mapping the *Berberis* reads with the assembled de novo reference sequences.

### 2.4. DEG

The transcriptome data sets of *B. thurnbergii* and *B. amurensis* were mapped against the transcriptomes of *B. koreana* for differentially expressed gene (DEG) analysis. Transcript read counts were calculated with StringTie v1.3.4d [38]. DEG analysis was performed using DESeq v1.30.0 [39]. In DESeq, read count is normalized through size factor and scatter plot, and significant DEG results are organized through Log2Fold Change value. Then, *p* value was calculated for post normalization in which we set *p*-value < 0.05 and log2FC abs1 for significance.

### 2.5. Functional Annotations and Phylogenetic Analysis

For functional annotation of genes obtained through differential expression analysis, a homology search was performed using a known protein database by Basic Local Alignment Search Tool (BLAST) analysis [40,41]. The cut-off parameters were E-value 10^−4^ and >70% similarity with the protein sequences in UniProt, TAIR, and NCBI databases. Finally, all candidate transcripts with Pfam domains and BLASTP hits were retained and imported to Blast 2GO suite 5.2 (BioBam Bioinformatics SL, Valencia, Spain). Then, gene ontology (GO) and Kyoto Encyclopedia of Genes and Genomes (KEGG) pathway analyses were performed. The retrieved GO terms were classified into three main categories: biological process (BP), cellular component (CC), and molecular function (MF). Unique enzyme commission (EC) numbers were assigned to transcripts and used to map the KEGG biochemical pathways. The specific composition of the GO terms was calculated and presented in a bar chart according to the percentage. 

Phylogenetic analysis was performed using the protein sequences of genes involved in berberine biosynthesis. The protein sequences were aligned with the same homologs of closely related families, and a maximum-likelihood phylogenetic tree was built using MEGA X (v.10.2.4) [42]. The protein sequences were aligned by ClustalW software (v2.1) and the phylogenetic tree was constructed by neighbor-joining method using MEGA ver. 5.0 software with 1000 bootstraps.

## 3. Results

### 3.1. Contents of Berberine and Berbamine in Leaf Tissues of Berberis Species 

HPLC analysis revealed that berbamine content was 20 times higher than berberine in leaf tissues of the three Berberis species (Figure 1). *B. thunbergii*, Japanese barberry, contained the highest content of berberine (19.2 ± 0.5 μg/g), followed *by B. amurensis* (16.5 ± 1.2 μg/g) and *B. koreana* (15.1 ± 1.9 μg/g) (Figure 1A). The same pattern was observed in berbamine such that the Japanese barberry was highest (421.6 ± 27.1 μg/g), followed by *B. amurensis* (314.3 ± 43.8 μg/g) and *B. koreana* (85.8 ± 4.8 μg/g) (Figure 1B).

### 3.2. mRNA Sequence Analysis and De Novo Assembly

Cellular mRNAs from leaf tissues were sequenced using Illumina paired-end sequencing technology. RNA-Seq libraries were made in triplicate in each sample. Raw reads were generated in each species in the range from 35 million to 80 million reads (Table 1). The average error rate of the libraries was as low as average Q20 > 96%. About 92% of the reads were retained in each species after removal of adaptor sequences, short reads (<50 nt), and low quality sequences (Table 1). Then, clean reads from all samples were pooled to perform de novo assembly using DRAP [32]. 

In the first assembly, the Oases generated 1.2 million contigs of 450 Mbp with an average length of 368.4 bp (Table 2), from which 1601 chimeras were removed. Thus, 65,258 open reading frames (ORFs) of 34 Mbp were obtained using TransDecoder (v3.0.1) after removing the redundant sequences (redundancy > 98%) using the CD-HIT program. Finally, BWA mapping reduced the number of obtained contigs to 42,278 of 44 Mbp with an average length of 1064.1 bp. The percent of mapped reads to the barberry reference genome assembly was 56.3% in *B. koreana*, 50.8% of *B. amurensis*, and 45.3% of *B. thunbergii* (Table 3). The unmapped reads could be attributed to either misassembled or absent sequences in the reference assembly. The total numbers of transcripts were 41,764, of which 39,209 were in *B. koreana*, 39,046 in *B. thunbergii,* and 38,214 in *B. amurensis.* The results are shown in rows P, Q, and R in Appendix A. 

### 3.3. Functional Annotations and Classifications 

For functional annotation of the transcripts, we queried the sequences of transcripts against public databases using BLASTX. As a result, the numbers of transcripts of *Berberis* species matched with the annotated transcripts in the NCBI database were 37,176 in Nr, 17,252 in GO, 4729 in KEGG, 24,364 in Pfam, and 23,466 in Interpro (Appendix A). GO terms using the Blast2Go program (BioBam Bioinformatics SL, Valencia, Spain) describe the annotated genes, gene products and sequences in controlled vocabularies into three subcategories; biological process (BP), cellular components (CC) and molecular function (MF). In *B. koreana* 20,806 were assigned to BP, 20,779 to CC and 20,821 to MF. The classification patterns of the other two *Berberis* species are similar to that of the *B. koreana* (Appendix A).

KEGG pathway analysis was carried out to seek the biological interpretations of the transcripts. Of the 41,464 unigenes, 4730 were able to be placed in 337 unique KEGG pathways in the three *Berberis* species (Appendix A). The number of annotated unigenes was 3804 in *B. koreana,* 3823 in *B. thunbergii,* and 3785 in *B. amurensis*. The unigenes were categorized into two major KEGG categories, Metabolism and Genetic Information Processing (Appendix A).

### 3.4. Differentially Expressed Genes and Classification

Differentially expressed genes (DEGs) among the *Berberis* species were identified by read counts using String Tie [38] and the results were normalized through size factor and scatter plot using DESeq [39] (Appendix A). Because transcripts of *B. koreana* were reported previously [30], we used the transcripts of *B. koreana* as a control in the comparison of gene expression levels of *B. amurensis* and *B. thunbergii.* Of the 41,464 unigenes, 37,404 were found to be differentially expressed among the three *Berberis* species. There were 50 in *B. koreana*, 1046 in *B. amurensis*, and 566 in *B. thunbergii* (Appendix A). Then, DEGs were further analyzed in GO categories. In GO categories, the numbers of differentially expressed transcripts were 3014 (473 upregulated and 2541 downregulated) in *B. amurensis* and 4494 (945 upregulated and 3540 downregulated) in *B. thunbergii* in comparison to those of *B. koreana,* respectively, in significance at *p* value 0.05 and log2FC > 1 (Table 4). *B. thunbergii* showed a higher number of unigenes differentially expressed compared to *B. amurensis* (Figure 2; Appendix A). 

In the KEGG annotations among the DEGs, a higher number of unigenes differentially expressed in *B. thunbergii* was observed compared to *B. amurensis* (Figure 3). In addition, the order of unigenes abundance in each pathway was different between the two species. For instance, the “pentose and glucoronate interconversions” pathway was enriched with the highest number of differentially expressed unigenes *B. thunbergii* (38 unigenes), but in *B. amurensis* the number was low (17 unigenes) (Figure 3). The exact numbers of upregulated and downregulated unigenes in both species are shown in Appendix A. 

### 3.5. Genes Encoding Enzymes for BIAs

Berberine biosynthesis begins with combining dopamine and 4-hydrophenylacetaldehyde, which are converted to *(S)*-norcoclaurine by *(S)*-norcoclaurine synthase (NCS) [30]. Then, *(S)*-norcoclaurine is converted to various BIA molecules by consequential steps that are mediated by several methyltransferases and oxidases. Table 5 shows the expression values (normalized read counts) of the unigenes of enzymes involved in berberine synthesis. The number of paralog copies of *B. koreana* are shown in brackets in the column of *B. koreana* from the results of Roy et al. [30], but these values are not shown in the *B. thunbergii* and *B. amurensis* because long-read cDNA transcriptomes are not available for them. Of the three species, *B. thunbergii* had highest berberine contents in leaf, followed by *B. amurensis* and *B. koreana* (Figure 1). However, the expressions of the berberine synthesis genes were not related to the berberine contents. For instance, expression values NMCH/CYP80B3 was in the order of *B. koreana*, *B. amurensis*, and *B. koreana*. There were eight steps from (*S*)-norcoclaurine to berberine in the berberine biosynthetic pathway [11]. Of the eight steps, four steps were mediated by methyltransferases, and four steps by oxidases. In the current study, unigenes encoding for those eight BIA biosynthesizing enzymes were all identified and expressions of these genes were differed among the three *Berberis* species (Figure 4A, Table 5, Appendix A). For instance, (*S*)-N-methylcoclaurine 3′-hydroxylase/N-methylcoclaurine 3′-monooxygenase (NMCH/CYP80B3), and 3′-hydroxy-N-methylcoclaurine 4′-O-methyltransferase (4-OMT) were found to be expressed highest in *B. amurensis*, while TyrAT (Tyrosine aminotransferase) and (*S*)-scoulerine 9-O-methyltransferase (SOMT) were expressed highest in *B. amurensis*. BBE (Berberine bridge enzyme/reticuline oxidase), canadine synthase enzyme (CYP719A21/CAS) and Tyrosine decarboxylase (TYDC) were expressed highest in *B. koreana* (Table 5). 

The berbamine biosynthetic pathway has not been fully elucidated yet. It was suggested that berbamunine was a precursor for berbamine [43]. Berbamunine is a molecule derived from (*S*)-N-methylcoclaurine by CYP80A1/berbamunine synthase [11]. We found a single copy of the CYP80A1 in the full length cDNA transcripts in *B. Koreana* [30]. 

### 3.6. CYPs and Methyltransferases among Three Berberis Species 

Cytochrome P450 mono-oxidases (CYPs) and methyltransferases are key enzymes in BIA synthesis. We identified 95 CYP gene families among the three species in our transcriptome data. 

The number of CYP copies was 257 in *B. koreana*, 255 in *B. thunbergii,* and 253 in *B. amurensis* (Appendix A). In the berberine biosynthesis pathway, CYP80B3 and CYP719A were involved, and we identified 11 copies in CYP80B3 and 6 copies in CYP719A of these CYPs in our transcriptome data (Appendix A). The CYP80A1 is responsible for the synthesis berbamunine to lead berbamine synthesis. There was a single copy of the CYP80A1 in *B. koreana*. From the DEG analysis, we observed that the unigenes expressing CYP719A were upregulated in *B. koreana* and downregulated in *B. thunbergii,* and those annotated as CYP80B3 were mostly downregulated in *B. koreana* but upregulated in *B. thunbergii* (Figure 4B). 

Our study annotated 298 unigenes as methyltransferases in 23 families (Appendix A). Of the 23 families, 4 families (4OMT, 6OMT, CNMT, and SOMT) are in the berberine synthesis pathway. The number of unigenes in 4OMT and SOMT were 2 in all three species. For 6OMT, the number of unigenes was 14, 13, and 15 in *B. koreana, B. thunbergii*, and *B. amurensis*, respectively. The number of unigenes of SOMT was 9, 10, and 10 in *B. koreana, B. thubergii*, and *B. amurensis*, respectively (Appendix A).

### 3.7. Phylogenetic Analysis of the Methyltransferases and CYP Oxidases

Protein sequences from the longest ORF of the four methyltransferases (6OMT, 4OMT, SOMT, and CNMT) and three CYPs (CYP719A, CYP80B3, and CYPA1) were employed for phylogenetic analysis (Figure 5). The methyltransferase tree was divided into two broad clusters, one with 6OMT and 4OMT and the other with SOMT and CNMT. Out of all *Berberis* methyltransferases, only SOMT was grouped together with orthologs of other species, appearing very close to *Thalictrum flavum* subsp *glaucum* and *Coptis chinensis* and *C. japonica* (Figure 5A). Other *Berberis* methyltransferase (6OMT, 4OMT, and CNMT) were tied with CNMTs in other species (Figure 5A). 

The CYP phylogenetic tree revealed two clusters, one for CYP80 and another for CYP719A (Figure 5B). In the CYP80 cluster, the *Berberis* CYP80A was out-grouped to other CYP80B subfamilies in which the two CYP80B from *Berberis* species formed a deep clade with *Thalictrum thalictroides*. In the CYP719 subfamilies, the *Berberis* CYP719 was out-grouped to other CYP719s from the genera *Thalictrum*, *Coptis*, and *Paperver*. 

## 4. Discussion

The species in the genus *Berberis* have been utilized for folk medicine for a long time worldwide. We carried out comparative transcriptome analyses on the three East Asian endemic *Berberis* species, *B. koreana, B. amurensis,* and *B. thunbergii,* to elucidate the genes involved in benzylisoquinoline alkaloids (BIAs) and bis-benzylisoquinoline alkaloid (bisBIA). Berberine and berbamine are major bioactive compounds in the *Berberis* species [3,12]. We quantified contents of the berberine and berbamine from leaf tissues of the three species. Berbamine contents were 20-fold higher than berberine in the three *Berberis* species. While the biosynthetic pathway for berberine was well characterized [16,30], the berbamine biosynthesis pathway has not been elucidated [43]. Total extracts from *Berberis* species have been used for a long time in folk medicine around the world and berberine was the main alkaloid that has been utilized in the pharmaceutical industry. The results from current study showed higher contents of berbamine than berberine. Thus, functional studies on the pharmacological effects of the berbamine in *Berberis* species are required. 

GO annotation was made in 17,252 unigenes, which is approximately 41% of the total 41,764 unigenes, whereas 4729 unigenes were annotated in the KEGG analysis, which was 11.32% of the total number. Our previous study in *B. koreana* annotated a similar number (16,756) of unigenes into GO classification, but a significantly higher number (12,362) of unigenes in KEGG categories [30]. The major difference can be attributed to the large number of novel unannotated unigenes found in the two related species *B. thunbergii* and *B. amurensis.* We identified approximately the same number of unigenes among the three species, but *B. thunbergii* showed a higher number of differentially expressed unigenes than *B. amurensis* in comparison with *B. koreana* in DEG analysis (Figure 2 and Figure 3). However, it would be too rash to relate the DEG results with the contents of berberine and berbamine because the expression of unigenes involved BIA synthesis was variable among the three species (Table 5). Prior to this study, we identified all genes encoding the enzymes in berberine biosynthesis in *B. koreana* and noted that branching steps to lead other alkaloids were effectively blocked by lacking the appropriate genes [30]. We demonstrated expression differences of the genes in berberine synthesis among the three *Berberis* species. Evolutionary pressures shape the gene expression by the relative importance of evolutionary changes in regulatory genetic and epigenetic mechanisms [44]. Thus, each *Berberis* species might have undergone different histories in adapting to local habitats. 

CYP450s are involved in wide array of plant metabolisms leading to the synthesis of fatty acids, lignin, plant hormones, and secondary metabolites [45]. Of the many CYP families, two main families were known to be involved in the BIA alkaloids: CYP80 and CYP719 [11]. We identified 95 CYP gene families, including CYP80 and CYP719 in the transcriptomes of the three species. Three CYP80 subfamilies (A, B, and G) were known in the BIA metabolism. CYP80B [(S)-N-methylcoclaurine 3′-hydroxylase. N-methylcoclaurine 4′-O-methyltransferase, NMCH] catalyzes the conversion of (S)-N- methylcoclaurine to (S)-3′ hydroxyl-N-methylcoclaurine leading to the berberine synthesis [42]. In our results, CYP80B3 showed highest level of expression counted by normalized read counts among the unigenes encoding berberine synthesis enzymes (Table 5). *B. amurensis* showed the highest level of expression, which was followed by *B. thunbergii* and *B. koreana*, which is an interesting result because the content of berberine in *B. thunbergii* was the highest among the three species. The CYP80G subfamily mediates the reaction of conversion of N-methylcoclaurine to lirinidine of aporphine in lotus [46]. The CYP80A (berbamunine synthase) subfamily mediates the reaction to convert the (*S*)-*N*- methylcoclaurine to berbamunine [43,47]. Berbamunine was the first dimer in the biosynthesis of bisBIAs caused by feeding the analysis of ^14^C-labeled tyrosine, tyramine, and several chiral 1-benzyl-1,2,3,4-tetrahydroisoquinolines in the cell culture of *Berberis stolonifera* [23]. Kraus and Kutchan (1995) [48] demonstrated this by heterologous expression of a cDNA encoding berbamunine synthase (CYP80) from a cell suspension culture of *B. stolonifera*, but they did not classify the CYP80 into subfamilies. In our results, the content of berbamine is about 20 times higher than berberine in the three *Berberis* species, but unigenes for encoding CYP80A/berbamunine synthase were not found in transcriptome data. Instead, we identified one copy of CYP80A/berbamunine synthase in the full-length cDNA data set in *B. koreana* which was generated by PacBio sequencing protocol [30]. In *Berberis* species, berbamine is a sole bisBIA alkaloids reported, whereas several BIA alkaloids (i.e., including berberine, palmatine, jatrorrhizine, tetrahydropalmatin, and berbamunine) were known [49], thus the expression level of the other enzyme genes in the berberine biosynthesis pathway might be higher than CYP80A/berbamunine synthase. 

There are two types of bisBIAs, noncyclic bisBIAs and cyclic bisBIAs. The berbamunine is a noncyclic BIA, whereas the berbamine is a cyclic BIA. The berbamunine may be an intermediate molecule leading to berbamine by forming an aryl bond between C8-O-C7 and C3′-O-C3′ in berbamunine. However, the enzyme in charge of the conversion berbamunine to berbamine is not known yet [43]. Although it is a prerequisite to quantify berbamunine in *B. koreana*, *B. thunbergii*, and *B. amurensis*, we could not identify the berbamunine in our samples in HPLC analysis because the standard berbamunine molecule was not available in our analysis. 

CYP719 subfamily members catalyze the formation of methylenedioxy bridge in roemerine (CYP719A1) lotus [9] and (*S*)-canadine (Canadine synthase, CYP719A21) in *Berberis* [30] and *Coptis* [21]. The expression levels CYP719A1 were much lower than expression levels of CYP80B3 in the three *Berberis* species in our results. We identified three copies of CYP719A1 in our previous study [30] and three more copies were identified in *B. koreana* in the current study. Thus, six copies of CYP719A1 were also present in *B. thunbergii* and *B. amurensis* (Appendix A). The number of copies of CYP719 in *Berberis* seemed to be higher than other BIA-producing plants, such as a single copy in lotus [9,50] and two copies in *C. deltoidea* [21]. The phylogenetic analysis divided the CYPs into two clusters, where *Berberis* CYP719A formed a sub cluster with *A. mexicana* and *P. somniferum. Berberis* CYP80 was formed into two sub-clusters where all other species formed one group, and *Berberis* CYP80 singly branched out. The *N. nucifera* appeared as an outgroup and a primordial sequence for both CYPs (CYP80B3 and CYP719A), which is in congruence with other studies [9,30].

We identified 24 methyltransferase families in which the norcoclaurine 6-*O*-methyltransferase (6OMT) had the highest number of copies with 13~15 copies, followed by *S*-coclaurine-*N*-methyltransferase (CNMT) with 9~10 copies in each *Berberis* species. The copies of these two transcripts in *Berberis* species were higher than the copies in other BIA producing species such as *C. deltoidea* with five copies of 6OMT and three copies of CNMT [21] and lotus with three copies of 6OMT and two copies of CNMT [9]. Methyltransferases catalyze biochemical reactions for biosynthesis and modifications of bioactive molecules in plants. There are several types of methyltransferases, depending on the substrate atom that accepts the methyl group; *O*, *N*, *C,* or *S* [8]. (*S*)-norcoclaurine, the most immediate BIA molecule, transformed to other BIA molecules by sequential methylation and oxygenation [16]. Methylation occurs at four sites at C6, C7, C4′, and N2 in benzylisoquinoline. In the berberine biosynthesis pathway, three *O*-methyltransferases and one *N*-methyltransferase are involved such as 6OMT, CNMT, 3′-hydroxy-*N*-methylcoclaurine 4′-*O*-methyltransferase (4OMT), (*S*)-scoulerine 7-*O*-methyltransferase (SOMT) [46]. Roy et al. (2021) [30] reported that the copy numbers of 6OMT, 4OMT, SOMT, and CNMT were two, two, one, and four, respectively, in *B. koreana* according to PacBio full-length cDNA analysis. In the current comparative transcriptome analysis, we identified additional copies to be 14 in 6OMT, 2 in 4OMT, 2 in SOMT, and 9 in CNMT in *B. koreana.* The copy numbers of 4OMT and SOMT were invariable among the three *Berberis* species, but 6OMT and CNMT were variable in numbers such that 6OMT was 14, 13, and 15 in *B. koreana, B thunbergii*, and *B. amurensis*, respectively. The copy numbers of CNMT were 9, 10, and 10 in *B. koreana*, *B thunbergii*, and *B. amurensis*, respectively. The phylogenetic analysis represented the *Berberis* methyltransferase (6OMT, 4OMT and CNMT) into a single clade together with CNMT of *Sinopodophyllum hexadendrum* and one sub cluster of four species (Figure 5A) indicating that the *Berberis* methyltransferase were not phylogenetically related to previously reported 6OMT and 4OMT genes. The current SOMT clade corroborated our previous findings [30] as it was tied closely to *T. thalictroides*, *C. japonica,* and *C. chinensis.*

In conclusion, the genus *Berberis* contains many medicinal plants in which the major bioactive molecules are bisbenzylisoquinoline (BIAs). Among the diverse BIAs, berberine is the main molecule that is the most biologically active BIA compound distributed in almost all *Berberis* species. Berbamine is a bisBIA molecule on *Berberis* species, but it has not been well studied yet compared to the wealth of information on berberine. We quantified both berberine and berbamine in three closely related *Berberis* species, *B. koreana*, *B. thunbergii*, and *B. amurensis*, which are endemic plants in Far East Asia. Unexpectedly, berbamine contents were almost 20 times higher than the berberine in all three species. This is the first report on the berbamine quantification compared to berberine in *Berberis* species. We identified all genes encoding the berberine synthesis from transcriptomes of the three species. For berbamine synthesis, we identified a single copy of the berbamunine synthase gene from PacBio sequencing database. Comparative genomic analyses were carried out on the transcriptomes of the three species, which revealed differential expression and copies of the BIA synthesis genes among the three species. Thus, the contents of the current research will be valuable for molecular characterization for the medicinal utilization of *Berberis* species. 

## Figures and Tables

**Figure 1 plants-11-02676-f001:**
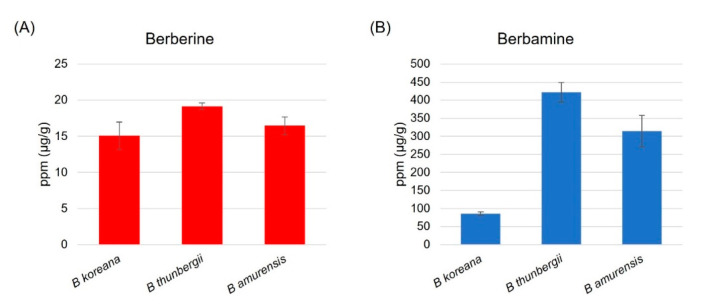
(**A**) Berberine and (**B**) Berbamine content in leaf tissue in the three species of *Berberis* based on the HPLC analysis; *B. thunbergii* has the highest berberine and berbamine content among the three.

**Figure 2 plants-11-02676-f002:**
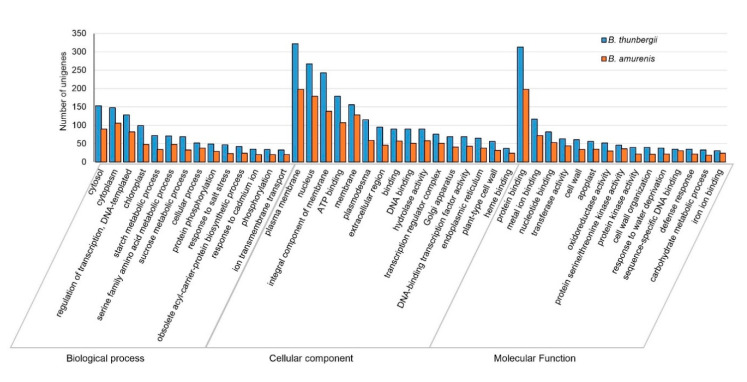
Gene Ontology (GO) enrichment analysis of differentially expressed unigenes in *B. thunbergii* (blue bars) and *B. amurensis* (orange bars) with *B. koreana* as control. The unigenes were annotated into distinguished groups named biological process, molecular function, and cellular component. The number of differentially expressed unigenes is higher in *B. thunbergii* compared to *B. amurensis*.

**Figure 3 plants-11-02676-f003:**
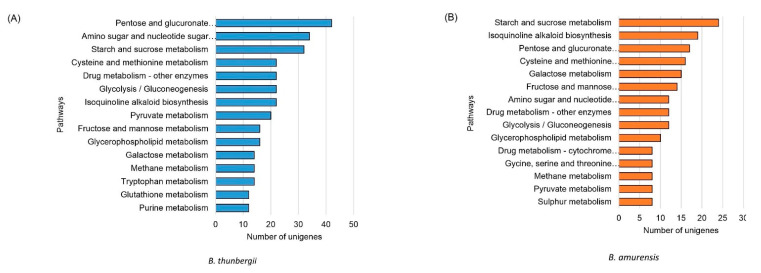
Functional annotation of differentially expressed unigenes in KEGG analysis in (**A**) *B. thunbergii* vs. *B. koreana* (blue bars) (**B**) *B. amurensis* vs. *B. koreana* (orange bars). *B. thunbergii* shows higher number of unigenes in each pathway compared to *B. amurensis*.

**Figure 4 plants-11-02676-f004:**
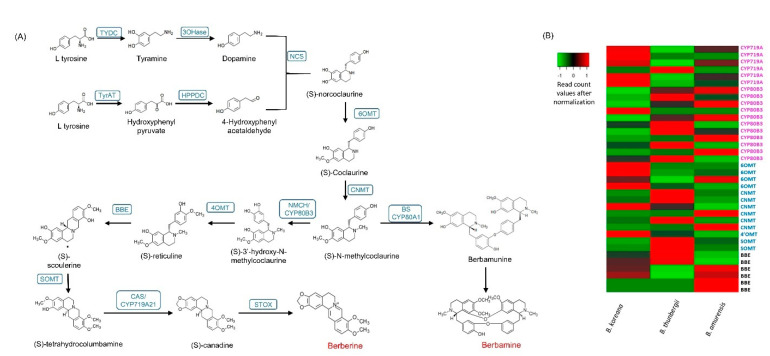
(**A**) Berberine biosynthesis pathways and (**B**) differentially expressed unigenes in three Berberis species. Blue letters represent enzyme names. Acronyms: TYDC (tyrosine decarboxylase); 3OHase tyrosine/tyramine 3-hydroxylase/tyrosine 3-monooxygenase; TyrAT (Tyrosine aminotransferase); NCS ((S)-norcoclaurine synthase); 6OMT ((S)-norcoclaurine 6-O-methyltransferase); CNMT ((S)-coclaurine N-methyltransferase); NMCH ((S)-N-methylcoclaurine 3′-hydroxylase, CYP82B subfamily); 4′OMT ((S)3′-hydroxy N methylcoclaurine 4′-O-methyltransferase); BBE berberine bridge enzyme (reticuline oxidase); CAS/Cyp719A21 (S)-canadine synthase; STOX (S)-tetrahydroprotoberberine oxidase; BS CYP80A1 berbamunine synthase.

**Figure 5 plants-11-02676-f005:**
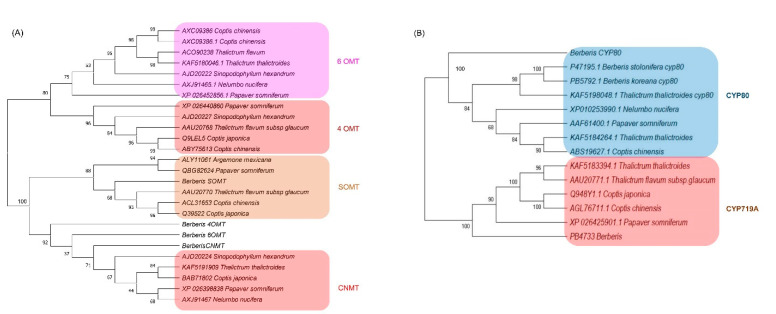
Phylogenetic analysis of (**A**) methyltransferases and (**B**) CYPs identified in *Berberis* species transcriptome. Numbers above branches show bootstrap percentages (BP). The protein sequences were aligned by ClustalW software and the phylogenetic tree was constructed by neighbor-joining method using MEGA ver. X software with 1000 bootstraps.

**Table 1 plants-11-02676-t001:** Raw data summary and statistics of transcriptome data in *Berberis* species.

	Raw Data			Filtered Data		
Species	Reads	Total Length	>Q20 (%)	Reads	Total Length	Rate (%)
*B. koreana*	35,998,374	5,435,754,474	97.41	33,381,816	4,733,258,684	92.7
*B. amurensis*	50,706,530	7,656,686,030	96.75	46,670,652	6,449,979,707	92
*B. thunbergii*	80,966,878	12,225,998,578	96.92	74,940,106	10,354,513,883	92.6

**Table 2 plants-11-02676-t002:** Overview of de novo assembly statistics.

	Oases (v0.2.09)	CD-Hit	Trans Decoder		Bwa
File	Oases	all_dgb	all_contigs	transdecoder.cds	coding_transcripts_fpkm
Total contigs	1,221,878	553,383	285,922	65,268	42,278
Total length	450,175,132	251,071,869	186,123,210	34,735,974	44,989,957
Min. length	100	50	200	297	244
Avg. length	368.4	453.7	651	532.2	1064.1
Max. length	4395	4395	5356	3627	5354

**Table 3 plants-11-02676-t003:** Mapping statistics in *Berberis* species.

Name	Total Reads	Mapped Reads	Mapped Rate (%)	Prop Paired Read	Prop Paired Mapped Rate (%)
*B. koreana*	33,381,816	18,809,373	56.3	16,617,264	49.8
*B. amurensis*	30,875,256	15,672,027	50.8	13,667,334	44.3
*B. thunbergii*	74,940,106	33,975,391	45.3	29,602,862	39.5

**Table 4 plants-11-02676-t004:** Differentially expressed genes of *Berberis* species in comparison.

Control	Target	Number of Transcripts Compared	*p*-Value < 0.05 & log2FC abs1		
			All	Upregulated	Downregulated
*B. koreana*	*B. amurensis*	39,521	3014	486	2528
*B. koreana*	*B. thunbergii*	40,036	4494	847	3334

**Table 5 plants-11-02676-t005:** Expression values of unigenes of enzymes involved in benzylisoquinoline alkaloid biosynthetic pathway in three *Berberis* species.

Enzyme	Abbreviations	*B. koreana* *	*B. thunbergii* *	*B. amurensis* *
Tyrosine decarboxylase	TYDC	425.10 (8) **	158.35	217.39
Tyrosine aminotransferase	TyrAT	1068.84 (4)	2015.10	1719.99
(*S*)-norcoclaurine synthase	NCS	1880.62 (8)	514.78	1341.78
(*RS*)-norcoclaurine 6-O-methyltransferase	6OMT	20.02 (2)	7.29	54.43
(*S*)-coclaurine N-methyltransferase	CNMT	268.97 (4)	226.20	109.22
(*S*)-N-methylcoclaurine 3′-hydroxylase/N-methylcoclaurine 3′-monooxygenase	NMCH/CYP80B3	1485.02 (6)	3959.62	4927.05
3′-hydroxy-N-methylcoclaurine 4′-O-methyltransferase	4′OMT	18.13 (2)	43.63	82.71
Berberine bridge enzyme (reticuline oxidase)	BBE	40.32 (1)	13.67	26.16
(*S*)-scoulerine 9-O-methyltransferase	SOMT	51.68 (1)	123.71	42.06
Canadine synthase enzyme	CYP719A21	851.83 (6)	343.40	589.24
Berbamunine synthase ***	CYP80A1	- (1)	-	-

* Values are normalized read count based on volcano plot of Appendix A. ** The numbers of brackets are the number of paralogs found in PacBio Transcriptome database (30). *** Expression level of berbamunine synthase was not analyzed due to its absence in the Illumina transcriptome data sets.

## Data Availability

All data were submitted to the National Center for Biotechnology Information (NCBI). The de novo assembly data was registered under SRA SUB10636706 with BioProject ID PRJNA814432. The raw RNA sequencing BioProject ID was PRJNA856352 for *B. amurensis*, PRJNA856402 for *B. koreana*, and PRJNA856417 for *B. thunbergii*.

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
