# Peer review of "Comparative Transcriptomics for Genes Related to Berberine and Berbamine Biosynthesis in Berberidaceae"

_plants, 2022, doi:10.3390/plants11202676_

Round 1
Reviewer 1 Report
see the attached file

Author Response
Response to the Reviewer’s comments •
Reviewer 1
- Line 46: and carotenoids carotenoids “carotenoids” is written 2 times.
- Corrected
- Lines 146-153: The character size is smaller than elsewhere.
- Corrected
- Line 147: The version of the CD-HIT-EST program should be given. • Line 158: The version of DESeq should be given.
- Versions added.
- Line 160: Change & to “and”
- Changed
- Line 178: The version of the ClustalW software should be given. • Lines 246-248 : It seems that the 2 sentences mean the same
- ClustalW: version added
- Line 246-248: sentence changed
Line 246-248 (old): B. thunbergii showed a higher number of unigenes differentially expressed compared to B. amurensis (Fig 2). The numbers of upregulated and downregulated unigenes were higher in B. thunbergii compared to B. amurensis (Supplementary Figs 4).
- Line 246-247: thunbergii showed a higher number of unigenes differentially expressed compared to B. amurensis (Fig 2; Supplementary Figs 4).

Reviewer 2 Report
This project carried out comparative transcriptome analyses on the three East Asian endemic Berberis species, B. koreana, B. amurensis and B. thunbergii to elucidate the genes involved in BIAs and bisBIA. Ninety-five CYP gene families, including CYP80 and CYP719 in the transcriptomes of the three species were identified, while the CYP80B3 showed highest level of expression among the unigenes encoding berberine synthesis enzymes. The expression levels of CYP719A1 were much lower than CYP80B3 in the three Berberis species. Moreover, 24 methyltransferase families,including two major members 6OMT and CNMT were identified. And the results indicated that the copy numbers and expression variation may relate to the diversitification of BIAs and bisBIA in different Berberis species. This is the first report on the berbamine quantification compared to berberine in Berberis species.The results of the current research will be valuable for molecular characterization for the medicinal uti-lization of Berberis species. But there are several points need to be concerned:
1) In Fig1, the standard error should be added.
2) The DEG genes in the key pathway should be validated by qPCR or other methods.
3) It’s better to carry out the primary functional analysis of the key genes in model species such as A. thaliana.
Author Response
Response to Reviewer 2
1) In Fig 1, the standard error should be added
-> Answer: Standard error bar added. Also standard errors are given in the text (see the lines 184 – 188).
2) The DEG genes in the key pathway should be validated by qPCR or other methods.
-> Answer: Thank you for your suggestion.
We used only the NGS RNA sequencing data to figure out the DEG concerning to the berberin and berbamine biosynthesis in this study. We agree to your comment that RT PCR test is a good evidence. However, NGS studies were proved to be enough to study DEG in other studies [Wang, et al. (2009) RNA-Seq: a revolutionary tool for transcriptomics. Nature Reviews Genetics. 10 (1): 57–63. doi:10.1038/nrg2484. PMC 2949280. PMID 19015660].
RT PCR may be slightly different from NGS data in our study. However, unfortunately we have no the fresh tissue to confirm it now.
3) It’s better to carry out the primary functional analysis of the key genes in model species such as A. thaliana
-> Thank you for your good suggestion. However, BIAs are specific alkaloids that are being produced in limited plant families. We commented this in the Introduction as “BIAs are restricted to certain plant families such as Magnoliaceae, Ranunculaceae, Papaveraceae, and Berberidaceae in Ranunculales [8] and Fabaceae in Fabales [9].” (see line 53 -54). The genes involved in the BIA synthesis were already proved their function in BIA producing plants (Hagel et al. 2011; BMC Plant Biol 2015, 15, 227, doi:10.1186/s12870-015-0596-0; Liu et al. 2017, Mol Plant 2017, 10, 975-989, doi:10.1016/j.molp.2017.05.007; He et al. (2018) Front Plant Sci 2018, 9, 731, doi:10.3389/fpls.2018.00731). These papers were cited in the Introduction in lines 66-67.
Moreover, the model plant A. thaliana does not synthesize the BIA alkaloids, thus the enzymes for BIA synthesis were not profiled in TAIR (TAIR - Home Page (arabidopsis.org).
